# Linguistic Embeddings as a Common-Sense Knowledge Repository: Challenges and Opportunities

## Abstract

Many applications of linguistic embedding models rely on their value as pre-trained inputs for end-to-end tasks such as dialog modeling, machine translation, or question answering. This position paper presents an alternate paradigm: Rather than using learned embeddings as input features, we instead treat them as a common-sense knowledge repository that can be queried via simple mathematical operations within the embedding space. We show how linear offsets can be used to (a) identify an object given its description, (b) discover relations of an object given its label, and (c) map free-form text to a set of action primitives. Our experiments provide a valuable proof of concept that language-informed common sense reasoning, or 'reasoning in the linguistic domain', lies within the grasp of the research community. In order to attain this goal, however, we must reconsider the way neural embedding models are typically trained an evaluated. To that end, we also identify three empirically-motivated evaluation metrics for use in the training of future embedding models.

## 1 Introduction

This position paper casts pre-trained embedding models like BERT (Devlin et al., 2018), GPT-2 (Radford et al.) and InferSent (Conneau et al., 2017) in a new role. Rather than using the learned hidden states as pre-trained features for downstream tasks, we instead view them as a form of emergent, flexible knowledge representation harvested from a rich body of text corpora. We show how the knowledge implicitly encoded in the embedding space can be extracted and utilized to solve real-world problems with little or no additional training, and we argue that the effectiveness of this method can be vastly increased in future embedding models.

There is, of course, inherent unpredictability of this approach. In contrast to hand-curated symbolic knowledge, harvested representational knowledge is imprecise, at times unreliable, and difficult to anticipate – but it is also fascinating, spontaneous, and fluidly creative in ways that traditional knowledge systems seldom replicate. Unlike a knowledge graph, these vector-based representations can explore questions like, "What is the combination of fear and sound?" or "What is like a river without water?[1]"

Queries, in this context, are linear algebra operations on the embedded representations of input text, with answers found by seeking the embedded word or sentence with the greatest cosign similarity to the calculated result. This requires a startlingly high level of semantic structure within the geometry of the learned representations, and gentle probing of current state-of-the-art embedding spaces reveals that they are not yet up to the task. The *potential* exists, but only in nascent form. The purpose of this paper is to unequivocally demonstrate that potential, quantify it where possible, and propose new evaluation metrics for the development of embedding models with optimal semantic geometries.

---

[1] If you query a FastText (Bojanowski et al., 2017) model, the answers are "scream" and "road".

## 2 RELATED WORK

Language-based knowledge representation falls roughly into two categories. The first includes knowledge graphs (Vrandecic & Krotzsch, 2014; Liu & Singh, 2004; Saxena et al., 2014), lexical databases (Miller, 1995), and structured ontologies (Matuszek et al., 2006). Each of these has distinct characteristics, but they share the principle of representing ideas as distinct entities linked by conceptual relations. Lexical knowledge may be hand-crafted (Liu & Singh, 2004; Miller, 1995; Matuszek et al., 2006) or extracted from the internet (Vrandecic & Krotzsch, 2014; Saxena et al., 2014), although almost all such extracted knowledge bases involve some form of human curation.

The second category is model-based, and involves allowing the system to learn a numerical representation for each word or phrase encountered. A neural network or statistical model is commonly used for this purpose, with the resulting representations interchangeably referred to as vector spaces (Mikolov et al., 2013a; Turney & Pantel, 2010), embedding spaces (Liu et al., 2015; Fulda et al., 2017a; Bolukbasi et al., 2016), or distributed representations of words (Le & Mikolov, 2014a; Mikolov et al., 2013b; Le & Mikolov, 2014b). While these models are heavily valued for their use as input features for downstream tasks, the information encoded within the embedding space is seldom *directly* extracted, particularly when the model encodes entire sentences or multi-word phrases.

One purpose of this paper is to raise a warning voice: In the mad scramble to train more clever and more effective embedding models, we should take note of our assumptions about what a 'good' embedding model looks like. One common evaluation method for neural language models is to measure their effectiveness as pre-trained input features for downstream tasks. Bert (Devlin et al., 2018), InferSent (Conneau et al., 2017) and Google's Universal Sentence Encoder (Cer et al., 2018a) were all evaluated using this method. However, while this evaluation method can reveal how well a system is learning language data in a general sense, it cannot tell us whether the resultant embedding space exhibits strong semantic structure. After all, a model trained to *predict* semantic similarity is not the same as a model whose sentence representations *embody* semantic similarity.

We therefore caution readers that currently popular evaluation metrics for linguistic embeddings may be overlooking some of the models' greatest potential strengths, and explicitly encourage efforts like those of Conneau et al. (Conneau et al., 2018) and zhu et al. (Zhu et al., 2018) to examine the semantic properties of learned representations directly. Otherwise, we may inadvertently bias our research agendas away from the very properties that make these embedding spaces so delightfully intriguing.

## 3 BACKGROUND: SEMANTIC PROPERTIES OF LEARNED REPRESENTATIONS

Word-level embedding spaces exhibit intriguing semantic geometries. For example, it is possible to perform vector operations within the embedding space to answer queries such as Spain:Madrid::France:?, where the unknown value *?* should be replaced by *Paris* (Mikolov et al., 2013b;c). Similarly, targeted projections of the embedding space onto hand-selected basis vectors reveal that words that describe common household objects are positioned in close proximity to objects that tend to appear within the same room. For example, common appliances like 'stove', 'refrigerator', and 'blender' lie close to one another and share a common angle of incidence toward the representation for 'kitchen' (see Figure 1).

These observations suggest that a form of everyday common-sense knowledge is implicitly encoded within the structure of the embedding space. It 'knows', in some sense, that Paris is a city in France, just as it 'knows' that an oven can be found in a kitchen. Further demonstrations of this principle include the successful use of tagged word2vec embeddings to reason about the affordances of physical objects (Fulda et al., 2017a) and to determine the best mode of travel for reaching vacation destinations (Fulda et al., 2017b).

Unfortunately, the structural properties of word-level embedding spaces fail to manifest effectively in their sentence-level extensions. For example, most state-of-the-art embedding models map negation pairs such as "I am a cat" and "I am not a cat" to nearly identical vectors (see Figure 2). The models were not optimized for lingusitic reasoning tasks, and their structural irregularities prevent them from being easily used in the same way that word embeddings are applied. Perhaps for this

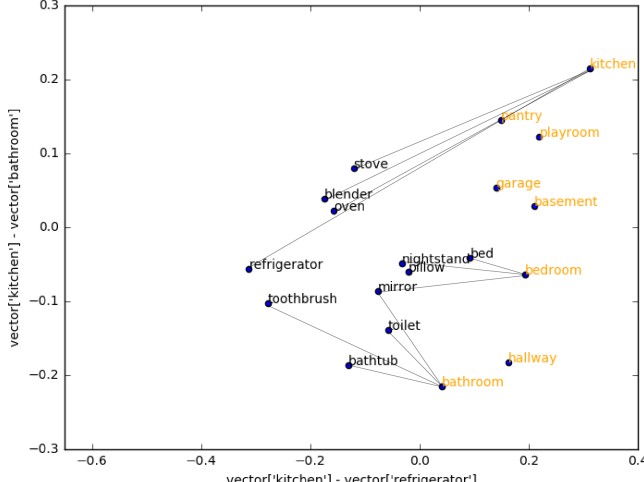

Figure 1: 100-dimensional word2vec embeddings projected into a 2-dimensional space demonstrate impressive semantic structure. Common household items are clustered in specific relationships to the rooms in which they tend to be located. The angle of correspondence is at least as significant as proximity. Of particular interest is the word 'mirror', which is aligned both with the items found in a bedroom and with the items found in a bathroom.

| | Skip-Thought | USE lite | BERT | GPT-2 | InferSent | Transformer-XL |
|---|---|---|---|---|---|---|
| I am a cat | | | | | | |
| I am not a cat | 0.069 | 0.069 | 0.070 | 0.0007 | **0.241** | **0.1110** |
| I am a cat | | | | | | |
| I am a domesticated cat | 0.139 | 0.098 | 0.188 | 0.0015 | **0.147** | **0.0925** |

Figure 2: Cosine distances between sentences embedded using various neural models. Each sentence pair comprises two rows, with distances shown after the second component sentence. Most embedding models place the syntactically similar but semantically distinct sentences 'I am a cat' and 'I am not a cat' startlingly close to one another. Only the recently-developed embedding models InferSent and Transformer-XL (bolded) embed these sentences to positions that are farther apart than the semantically similar sentences 'I am a cat' and 'I am a domesticated cat'.

reason, many applications of pre-trained embeddings focus on their potential for transfer learning rather than on attempts to extract information directly from the embedded representation.

A major contribution of this paper is the idea that common-sense knowledge *can* be extracted directly from multi-word linguistic embeddings, and that this knowledge can be utilized for real-world applications. This is partially due to the improved structural properties of recently-developed embedding spaces, partially due to an extraction mechanism that focuses on the relative positions of sentences rather than on their absolute locations, and partially due to the careful selection of reasoning tasks that are compatible with the learned geometries of modern embedding spaces.

## 4 INDEXING: IDENTIFYING AN OBJECT FROM ITS DESCRIPTION

When dealing with knowledge systems, a first and critical step involves identifying an entry point: In other words, which node should the system examine in order to access knowledge relevant to its current situation? In this section, we show that linguistic embeddings can be used to identify the object referred to by a free-form description, often with surprising accuracy. A key observation here is that different models not only perform differently, but they each answer a different subset of queries correctly. This suggests (but certainly does not prove) that a particularly well-formulated embedding space might facilitate much higher accuracies, if such an embedding space can be properly trained.

|  | Skip-Thought | USE lite | BERT | GPT-2 | InferSent | Transf-XL |
|---|---|---|---|---|---|---|
| a two-door car with a trunk and a solid roof | **coupe** | minivan | convertible | minivan | **coupe** | convertible |
| a car with a retractable roof | coupe | **convertible** | **convertible** | minivan | coupe | **convertible** |
| a family car with sliding doors and a large cargo area | coupe | **minivan** | suv | hollywood | **suv** | convertible |
| has more ground clearance for offroad travel | coupe | **suv** | **suv** | hollywood | **suv** | **suv** |
| that city that's always getting destroyed by monsters | tortoise | chicago | suv | hollywood | hollywood | suv |
| the place where all the movies are made | hamburg | **hollywood** | suv | **hollywood** | **hollywood** | suv |
| a port city in Germany | tokyo | **hamburg** | **hamburg** | duck | **hamburg** | suv |
| the windy city | coupe | **chicago** | **chicago** | holywood | **chicago** | suv |
| a turtle that walks on land | **tortoise** | **tortoise** | suv | duck | **tortoise** | suv |
| a water bird with a squat beak that quacks | tortoise | **duck** | **duck** | kitten | **duck** | suv |
| a cross between a horse and a donkey | tortoise | minivan | **mule** | duck | **mule** | **mule** |
| an immature cat | tortoise | **kitten** | suv | minivan | **kitten** | **kitten** |
| Total | 2 | 9 | 6 | 1 | 9 | 4 |

Figure 4: Indexing labels chosen by six neural embedding models when applying the methodology described in Section 4.1. Correct answers are shown in bold-face type. Interestingly, the best performing models tend to select a label in the right general category even when they do not select the optimal answer.

## 4.1 METHODOLOGY

Given a free-form text description $x$, a set of possible node labels $L = \{l_1, ..., l_n\}$, and two sets of example texts $A = \{A_1, ..., A_n\}$ and $B = \{B_1, ..., B_n\}$, where each $A_i$ is a free-form description of some object, and each $B_i$ is the object so described, a linguistic embedding model can be queried for common-sense knowledge in the following manner:

1. Use the model to convert $x$ into a vector representation $v$
2. Create a canonical vector $V = \frac{1}{n} \sum_i (B_i - A_i)$
3. Apply the canonical vector to obtain an indexing point $p = v + V$
4. Search the set of possible node labels to find $l^* = argmin_{l \in L} \, dist_{cos}(l,p)$

Here, $dist_{cos}$ is the cosine distance between two vectors.

## 4.2 INDEXING CASE STUDY

Figure 4 shows the indexes selected by six neural embedding models when given the texts in Figure 3 as canonical examples and twelve node labels to choose from.

We begin by noting that we are using these embedding models in ways for which they were not designed. GPT-2 and Transformer-XL, in particular, were trained not as general purpose sentence representations but for the very specific purpose of language modeling and text generation. It is perhaps not surprising, then, that they are not well suited to the task of node indexing via vector offset methods. Nevertheless we include them in this study because it is enlightening to observe that different training corpora and model architectures create a startling amount of variance in the semantic geometry of the resultant embedding space.

| object | description |
|---|---|
| parrot | a brighly colored tropical bird that can learn to speak |
| puppy | a baby dog |
| sedan | a car with four doors and a traditional trunk |
| truck | a vehicle with an open cargo bed in the rear |
| paris | the city where the eifel tower is |
| london | capital city of great britain |

Figure 3: Object labels $A_i$ and descriptions $B_i$ that define the canonical vector $V$ used in the experiments for Figure 4.

While a case study of this size is far from conclusive, the results tell an interesting story. Of the embedding models examined, Google's Universal Sentence Encoder (lite) and Facebook's InferSent model clearly have the most effective geometries for this task. Both models were able to index 75% of the free-form text descriptions correctly, and even when they selected an incorrect label, they usually selected it from the right general category of answers. We find this impressive and evocative of future possibilities. The question quickly arises: If models that were trained without this specific application in mind can accomplish an indexing task with reasonable accuracy, what might a model accomplish if it were trained with the *specific objective* of facilitating knowledge retrieval via vector offset methods? Further thoughts in this direction are outlined in Section 7.

| | Skip-Thought | USE lite | BERT | GPT-2 | InferSent | Transf-XL |
|---|---|---|---|---|---|---|
| Donald Trump | **actor** | **politician** | **politician** | pay | **politician** | **actor** |
| Taylor Swift | **actor** | politician | *actress* | jack | **actor** | **actor** |
| cocker spaniel | politician | habitat | actor | olympic | **dog** | animals |
| parakeet | indigenous | birds | actress | outcome | *bird* | extinct |
| The Daily Planet | **newspaper** | **newspaper** | **newspaper** | **newspaper** | **newspaper** | **newspaper** |
| The New York Tribune | **newspaper** | **newspaper** | **newspaper** | **newspaper** | **newspaper** | **newspaper** |
| Topeka | **city** | country | **city** | \xd7 | **city** | charter |
| Vienna | **city** | **city** | **city** | egyptian | **city** | **city** |
| New Zealand | politician | **country** | auckland | newspaper | zealand | **country** |
| Ivory Coast | actor | city | city | european | city | city |
| Total | 6 | 5 | 5 | 2 | 7 | 6 |

Figure 6: Relation extraction. Given a starting node label, vector offsets were used to identify relations that fall loosely under the 'is-a' umbrella, such as 'occupation', 'instance-of' and subclass-of'. Relations that could be verified in WikiData are bolded. Italicized orange text indicates relations that are not reflected in WikiData, but that perhaps should be.

## 5 EXTRAPOLATION: DISCOVERING RELATIONS

In this section we demonstrate that, given only a few exemplars of a relation, linguistic embedding models are able to generalize that relation to new starting nodes. Figure 5 shows the canonical relation examples used for this task: Three public figures, one newspaper, one city, one type of animal, and one country name, each of which is paired with an object that fulfills the relation $A_i$ is-a $B_i$. We use this data to extrapolate new relations for ten previously unseen starting nodes in the following manner.

| label | relation |
|---|---|
| Tom Cruise | actor |
| Bill Gates | entrepreneur |
| Hillary Clinton | politician |
| The Daily Herald | newspaper |
| Paris | city |
| kitten | mammal |
| The United States of America | country |

Figure 5: Example texts $A_i$ and $B_i$ used to define the vector $V$ between a description and the corresponding object

Given an input node label $x$, a set of possible output nodes $L = \{l_1, ..., l_n\}$, and a set of example pairs $(A_i, B_i)$, where each $A_i$ is a node label and each $B_i$ is the label of an object that fulfills the desired relation, extrapolations to new input nodes can be calculated in the following manner:

1. Create a set of canonical vectors $V_i = B_i - A_i$
2. For each previously unseen node label $x$, convert $x$ into a vector representation $v$
3. Find the indexing point $p = v + V_{nearest}$, where $V_{nearest}$ is the canonical vector whose initial point $A_i$ has the closest cosign distance to $v$
4. Search the set of possible output nodes to find $l^* = argmin_{linL} \text{dist}_{cos}(l_i, p)$

Note that, when indexing as per Section 4.2, we calculated a single canonical vector $V$ based on all of the exemplar $(A_i, B_i)$ pairs. When finding relations, however, we seek to match the given exemplar relations as closely as possible. Thus we use a suite of vectors $V_i$, selecting the one whose source word $A_i$ is most similar to the input node vector $v$. This relational vector $V_i$ is then used to query the knowledge base.

Case study results are shown in Figure 6. The ten input node labels we used were chosen before any of the results had been calculated, and were not changed afterward. The output label set $L$ consisted of the 5000 most common words in the FastText (Bojanowski et al., 2017) token set.

For the purposes of this study, we defined a valid result as a $l^*$ label that fulfils a WikiData (Vrandečić & Krötzsch, 2014) relation of the type 'instance-of', 'subclass of', or 'occupation' (all of which can be considered a variant of the 'is-a' relation exemplified in Figure 5) with respect to the input node. The answer 'newspaper' was accepted as being equivalent to the WikiData node 'fictional newspaper', which was not present in our label set $L$.

Five of the six embedding models produced a valid $l^*$ on at least 5 or our 10 case study words. The InferSent and Transformer-XL models seemed particularly adept at representing knowledge in this way. Particularly interesting are the labels shown in orange italicized text. These are terms that do *not* appear in the WikiData results for each source node, but which are arguably correct anyway.

| | InferSent$_1$ | InferSent$_2$ | InferSent$_3$ | InferSent$_4$ |
|---|---|---|---|---|
| Donald Trump | **politician** | *businessman* | *citizen* | *politicians* |
| Taylor Swift | **actor** | *actress* | *actors* | **singer** |
| cocker spaniel | **dog** | *dogs* | *animal* | wildlife |
| parakeet | *bird* | species | *animal* | *birds* |
| The Daily Planet | **newspaper** | *newspapers* | magazine | web |
| The New York Tribune | **newspaper** | *newspapers* | magazine | web |
| Topeka | **city** | *town* | county | township |
| Vienna | **city** | *town* | borough | *municipality* |
| New Zealand | zealand | city | township | county |
| Ivory Coast | city | coastal | coast | municipality |

Figure 7: Relation extraction using the InferSent embedding model, showing the first four query results. Bolded text indicates relations that exist in WikiData. Orange italicized text indicates relations that are not present in WikiData, but that are arguably correct. Green italicized text indicates plural versions of valid relations found in WikiData.

This demonstrates the powerful potential of linguistic embedding spaces to extract relations which are lacking from more traditional knowledge representations.

To further the case study, we took a closer look at the top four label results given by the InferSent embedding model for each of our source nodes (see Figure 7). Source nodes are listed on the left; labels are listed left to right in order of increasing distance from the indexing point $p$. Examination of the extracted labels reveals innovative and potentially useful information that would not necessarily be encoded in a traditional hand-curated knowledge system: For example, that the President of the United States is also a *citizen*, or that a cocker spaniel is an *animal* as well as a *dog*. Taylor Swift is correctly identified as both an actor and a singer, while the results for Vienna include the insightful result that it is a *municipality* as well as a city.

## 6 LANGUAGE IN THE WILD: MAPPING HIGH-LEVEL COMMANDS TO ACTION PRIMITIVES

One way to test the amount of common-sense knowledge intrinsically present in a language representation is to see whether it can facilitate natural language reasoning: For example, given a set of action primitives $A = \{a_1, ... a_n\}$, a set of action labels $L$, and a mapping $\mathcal{M}(L) \to A$, the question arises: Can the geometric structure of the embedding space be used to map incoming commands from a human user to the correct action primitives?

To explore this question, we created a set of 58 user directives and 9 action primitive labels for a hypothetical scenario in which a human controller is giving voice commands to an unmanned aerial vehicle (UAV). Based on the user directive $u$, the UAV must determine which behavior to execute. This is done in the following manner:

1. For each human utterance $u$ received:
    (a) Convert the $u$ into a vector representation $v$
    (b) Find the best-matching action label $l^* = argmin_{l \in L}$ dist$_{cos}(l,v)$
    (c) Execute the motor primitive $\mathcal{M}(l)$

Results are shown in Figure 9. We begin by observing that none of the embedding spaces is particularly good at this task, however, all of them are able to outperform a random baseline. This indicates that a significant amount of common-sense knowledge is implicitly encoded within the sentence representations. The key question is: How do we create embedding spaces in which this common-sense knowledge is more effectively structured?

We begin by noting that the InferSent model, which was trained using a recurrent neural architecture that was optimized for only a single training task, has markedly better performance than models which trained on many tasks simultaneously. The second-highest performance was attained by the Skip-Thought architecture, which was also trained exclusively on a single task (Kiros et al., 2015). This suggests that the current trend of training powerful architectures on as many tasks as possible,

| human directive | action label |
|---|---|
| 'advance three feet' | *go forward* |
| 'forward, please' | *go forward* |
| 'increase altitude' | *go higher* |
| 'try to touch the ceiling' | *go higher* |
| 'hold still' | *stop* |
| 'maintain position' | *stop* |

Figure 8: **Left:** Example directives and action labels used in the UAV control task. **Right:** Illustration of the UAV control task. Sentence represenations are used to map user directives to action labels, which in turn can be mapped to direct motor controls.

| Skip-Thought | 34.48% |
|---|---|
| USE lite | 29.31% |
| BERT | 22.41% |
| GPT-2 | 15.52% |
| InferSent | 48.28% |
| Transf-XL | 31.03% |
| random | 11.11% |

Figure 9: Classification accuracy on a UAV control task in which human utterances are mapped to the most similar action label based on cosign distances within the embedding space.

while undeniably effective at improving performance at transfer learning (Devlin et al., 2018; Cer et al., 2018b), may not be the best way to produce sentence representations with strong internal semantic structures.

## 7    EVALUATION METRICS FOR REPRESENTATIONAL KNOWLEDGE BASES

When evaluating linguistic embedding models, researchers often evaluate model performance based on cross-task generalization. In this paradigm, the learned representations are used as input features for a small, often single-layer, network that leverages pre-trained structure. Thus the quality of the embedding model is implicitly defined in terms its facilitation of transfer learning, a property which appears strongly correlated with the use of multiple simultaneous training tasks.

There is of course, nothing inherently wrong with this approach. But if we wish to train embedding models for specific and targeted use as a common-sense knowledge repository, then transfer learning becomes less relevant. Instead, we choose to focus on the semantic geometries of the learned embedding space.

In this section, we propose three properties that seem correlated with high performance on the case studies in this paper. We observe that these desired properties can either be induced implicitly, by selecting learning tasks which can only be mastered when the desired embedding properties are present, or explicitly, by directly incorporating a measurement of the desired property within the loss function of the neural network.

1. *Analogical coherence.* The analogical properties observed in word2vec, GLoVE, FastText, and other single-word embedding spaces should be preserved in the trained sentence-level vector space such that offset relationships like Spain:Madrid::France:Paris are preserved. At the phrase or sentence level, this should extend to relationships like 'if you drop a ball':'it will bounce'::'if you drop a glass':'it will break'.

2. *Semantic alignment.* Single-word embeddings should be located closer to sentences that contain those words than to conceptually equivalent sentences that do not contain them. Sentences that express similar ideas should be located near one another despite variations in syntax or structural complexity, and arbitrarily-ordered "bags of words" should be located close to sentences in which those words appear.

|  | negation detection | clause relatedness | Google analogy |
|---|---|---|---|
| Skip-thought | 61.48% | 20.53% | 50.86% |
| USE lite | 77.78% | 2.48% | 52.12% |
| BERT | 89.48% | 6.02% | 46.56% |
| GPT-2 | 61.19% | 38.23% | 6.47% |
| InferSent | **97.48%** | **48.50%** | **81.81%** |
| Transf-XL | 26.37% | 17.88% | 47.08% |

Figure 10: Classification accuracy on three tasks related to semantic geometry. The highest value in each column is bolded.

3. *Polarity displacement.* A sentence and its negation should be located far from each other along at least one basis dimension of the space, and the negated sentence should be located close to non-negations that nevertheless convey the same concept. (For example, 'The room is not empty' should be located farther from the statement 'The room is empty' than it is from sentences such as 'The room is occupied' or 'The room is full'.)

The evaluation framework introduced by zhu et al. (Zhu et al., 2018) includes metrics that are relevant to two of these properties. Semantic alignment is measured by a clause relatedness metric that requires the distance between a sentence and one of its embedded clauses to be less than the distance between the original sentence and its not-negation, while polarity displacement is measured by the *negation detection* task. The third property, analogical coherence, can be effectively measured using the Google Analogy Test Set introduced by Mikolov et al. (Mikolov et al., 2013a).

Figure 10 shows the performance of four state-of-the-art embedding models on these tasks. The InferSent model outperforms the other models at all three tasks, just as it showed the highest performance on our (admittedly small) indexing, relation-extraction, and UAV control case studies. Even more interestingly, the only other model to score greater than 20% at all three tasks (Skip-Thought) was also the second-highest scoring algorithm on the UAV control task and tied for second on the relation extraction task (although it failed miserably on the indexing case study).

Although more extensive studies remain to be done, it seems to be the case that high performance on the evaluation tasks in 10 is correlated with strong performance in at least some applications of the embedding model as a common-sense knowledge base.

## 8 CONCLUSION

As the search for more efficient and effective embedding spaces progresses, it is imperative that researchers think critically, not just about the performance of the embedding space on established benchmarks, but on the potential of linguistic embedding spaces for charting new territory.

In this paper, we have examined some of the capacities (and weaknesses) of current sentence-level embedding spaces, and we have proposed a set of criteria that may help to guide the development of future embedding spaces. In particular, we have re-examined basic industry assumptions about how these embedding spaces are created, how they can be manipulated, and for which applications they are best suited. If we wish to extend analogical reasoning beyond elementary word vector calculations, we need a new kind of embedding space, one that functions not merely as useful input features for downstream tasks, but rather as a valuable common-sense repository in its own right: A knowledge base that can be queried mathematically to solve real-world problems with little or no direct training. In order to realize this potential, however, new embedding models must be developed which have been optimized for such applications.

We have proposed three broad training criteria - Analogical coherence, Semantic alignment, and Polarity displacement - which show signs of correlation with performance on knowledge extraction tasks, and have identified evaluation datasets within the existing literature that measure the extent to which learned sentence representations satisfy these criteria. We hope that, in addition to continued efforts to optimize linguistic embedding models based on transfer learning, researchers in the field will begin to explore the training and application of embedding models as common-sense knowlede repositories.

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
