# OpenReview forum: "Linguistic Embeddings as a Common-Sense Knowledge Repository: Challenges and Opportunities"
_ICLR.cc/2020/Conference — Reject_

### Official Review · AnonReviewer3 · 2019-10-23
**Official Blind Review #3**

**Rating:** 1

**Review:**

This paper provides an overview of methods to use embeddings for texts as common-sense knowledge. It mentions many aspects where embeddings can be used as common-sense knowledge. However, the paper lacks both novelty and in-depth analysis.  Most methods proposed are basically computing the cosine distances between language embeddings pairs and find the closest one. It's hard to imagine how to scale up this process in real applications. And most of the analyses are based on cherry-picked examples rather than evaluation with quantitative metrics.

**Experience Assessment:**

I have read many papers in this area.

**Review Assessment: Checking Correctness Of Derivations And Theory:**

N/A

**Review Assessment: Checking Correctness Of Experiments:**

I carefully checked the experiments.

**Review Assessment: Thoroughness In Paper Reading:**

I read the paper thoroughly.

---

### Official Review · AnonReviewer1 · 2019-10-24
**Official Blind Review #1**

**Rating:** 1

**Review:**

- This paper claims to provide a “alternate” view of pretrained embedding as commonsense repository. But this is not a new view at all. It is well known embedding can encode commonsense knowledge, which is a reason it helps a wide variety of recent commonsense related tasks (and there has been considerable analysis on what kind of coomonsense is learned and helpful).

- The paper also claims to propose three training criteria, which are not new as well. In fact, much work has been performed to learn representation for different semantic orientation, dimensions and relations (e.g., polarity like full/empty is one special case of contrasting meaning, which has been studied a lot in the distributed representation paradigm for years).

- The paper has not been carefully written yet; e.g., even in the abstract, there is a typo like “typically trained an evaluated”.

I do not think this paper, claimed as a position paper, adds any new positions to the existing literature. I do not recommend it for the conference.


**Experience Assessment:**

I have published in this field for several years.

**Review Assessment: Checking Correctness Of Derivations And Theory:**

I assessed the sensibility of the derivations and theory.

**Review Assessment: Checking Correctness Of Experiments:**

I assessed the sensibility of the experiments.

**Review Assessment: Thoroughness In Paper Reading:**

I read the paper at least twice and used my best judgement in assessing the paper.

---

### Official Review · AnonReviewer2 · 2019-10-26
**Official Blind Review #2**

**Rating:** 3

**Review:**

This is a position paper - it discusses overall about multi-word embedding (sentence) level and how it can be leveraged in multiple applications. Also, it discusses some open opportunities on how the embeddings can be used as a base knowledge source and some challenges in them

Pros:
1. The examples and analysis of embeddings involved in the paper is detailed
2. The authors have run lots of experiments to position their idea correctly.

Cons:
1. I find many of their intuitions well established in the literature - the intuition of geometric structures in the embedding space, the intuition of incidence angle. The ideas from section 4,5,6 seem to be rather obvious extension of what we know

2. Especially, in Section 4,5 the idea of finding indexes, is nothing but the projection of x on the mean of (A,B) - which is more of a common practice. I find both these ideas of finding indexes and extrapolation, pretty much recurrent and existing in literature.

3. What I would be really interested is, a new training algorithm for learning embeddings, that would be able to answer all linear algebra based questions in the embedding space. Rather, this paper analyses the pros and cons of 6 existing models. In my personal opinion, this is half work done for a very good paper.

4. Personally, I am not very appreciative of the use of  "Common Sense" in this paper. There is a set of papers in the literature (atleast in vision domain), where they actually try to learn the common sense from the underlying data. The use of common sense in this paper does not align with that usage and hence, is misleading for me.

5. From reading a position paper, I would ideally want to see ideas worth of atleast 2-3 PhD thesis. I do not find such compelling ideas in this paper.

**Experience Assessment:**

I have read many papers in this area.

**Review Assessment: Checking Correctness Of Derivations And Theory:**

N/A

**Review Assessment: Checking Correctness Of Experiments:**

I carefully checked the experiments.

**Review Assessment: Thoroughness In Paper Reading:**

I read the paper at least twice and used my best judgement in assessing the paper.

---

### Decision · Program_Chairs · 2019-12-19

**Decision:**

Reject

**Comment:**

This paper presents an analysis of the kind of knowledge captured by pre-trained word embeddings. The authors show various kinds of properties like relation between entities and their description, mapping high-level commands to discrete commands etc. The problem with the paper is that almost all of the properties shown in this work has already been established in existing literature. In fact, the methods presented here are the baseline algorithms to the identification of different properties presented in the paper.

The term common-sense which is used often in the paper is mischaracterized. In NLP literature, common-sense is something that is implicitly understood by humans but which is not really captured by language. For example, going to a movie means you need parking is something that is well-understood by humans but is not implied by the language of going to the movie. The phenomenon described by the authors is general language processing.

Towards the end the evaluation criteria for embedding proposed is also a well-established concept, its just that these metrics are not part of the training mechanism as yet. So if the contribution was on showing how those metrics can be integrated in training the embeddings, that would be a great contribution.

I agree with the reviewer's critics and recommend a rejection as of now.